# Towards a Common Definition for the Diagnosis of Iron Deficiency in Chronic Inflammatory Diseases

**DOI:** 10.3390/nu14051039

**Published:** 2022-02-28

**Authors:** Patrice Cacoub, Gabriel Choukroun, Alain Cohen-Solal, Elisabeth Luporsi, Laurent Peyrin-Biroulet, Katell Peoc’h, Valérie Andrieu, Sigismond Lasocki, Hervé Puy, Jean-Noël Trochu

**Affiliations:** 1Department of Internal Medicine and Clinical Immunology, Groupe Hospitalier Pitié-Salpêtrière, AP-HP, 75013 Paris, France; 2UMR S 959, Immunology-Immunopathology-Immunotherapy (I3), INSERM, UPMC Univ Paris 06, Sorbonne Universités, 75013 Paris, France; 3Biotherapy (CIC-BTi) and Inflammation-Immunopathology-Biotherapy Department (DHU i2B), Hôpital Pitié-Salpêtrière, AP-HP, 75013 Paris, France; 4MP3CV Laboratory, EA7517, Jules Verne University of Picardie, 80054 Amiens, France; choukroun.gabriel@chu-amiens.fr; 5Division of Nephrology, Amiens University Hospital, 80054 Amiens, France; 6UMR-S 942 MASCOT, Paris University, Lariboisière Hospital, AP-HP, 75010 Paris, France; alain.cohen-solal@inserm.fr; 7Oncology Unit, Mercy Hospital, 57530 Ars-Laquenexy, France; e.luporsi@chr-metz-thionville.fr; 8NGERE U1256, INSERM, Department of Gastroenterology, University Hospital of Nancy, University of Lorraine, 54500 Vandoeuvre-lès-Nancy, France; peyrinbiroulet@gmail.com; 9Department of Clinical Biochemistry, Beaujon Hospital, AP-HP, 92110 Clichy, France; katell.peoch@aphp.fr (K.P.); herve.puy@aphp.fr (H.P.); 10UMR 1149, Centre de Recherche sur l’Inflammation (CRI), INSERM, University of Paris, 75018 Paris, France; 11Department of Hematology, Bichat-Claude Bernard Hospital, AP-HP, 75018 Paris, France; valerie.andrieu@aphp.fr; 12Department of Anesthesia and Intensive Care Medicine, University Hospital of Angers, 49933 Angers, France; sigismond@lasocki.com; 13Institut du Thorax, INSERM, CNRS, University Hospital of Nantes, University of Nantes, 44093 Nantes, France; jeannoel.trochu@chu-nantes.fr

**Keywords:** iron deficiency, ferritin, TSAT, chronic inflammatory disease, definition, epidemiology

## Abstract

Iron deficiency (ID) in patients with chronic inflammatory diseases is frequent. However, under-diagnosis is also frequent due to the heterogeneity between guidelines from different medical societies. We applied a common definition for the diagnosis of ID to a large panel of patients with cancer, heart failure (HF), inflammatory bowel disease (IBD), and chronic kidney disease (CKD), where ID was defined as serum ferritin concentration <100 μg/L and/or a transferrin saturation (TSAT) index <20%. Prevalence estimates using this common definition were compared with that obtained with officially accepted definitions (ESMO 2018, ESC 2016, ECCO 2015, and ERBP 2013). For that purpose, we used data collected during the French CARENFER studies, which included 1232, 1733, 1090, and 1245 patients with cancer, HF, IBD, and CKD, respectively. When applying the common definition, ID prevalence increased to 58.1% (vs. 57.9%), 62.8% (49.6%), and 61.2% (23.7%) in cancer, HF, and IBD patients, respectively. Both prevalence estimates were similar (47.1%) in CKD patients. Based on our results, we recommend combining both ferritin concentration and TSAT index to define ID in patients with chronic inflammatory diseases. In those patients, adopting this common definition of ID should contribute to a better screening for ID, whatever the condition.

## 1. Introduction

Iron deficiency (ID) is the most common and widespread deficiency in the world, with more than 1.2 billion affected individuals with anemia, and probably more than double that without anemia [1]. Patients with chronic inflammatory diseases such as cancer, heart failure (HF), chronic kidney disease (CKD), and inflammatory bowel disease (IBD) are among the populations with the highest risk of ID [2]. In these patients, ID prevalence has been reported as high as 60–90% [2,3,4,5]. In patients with chronic inflammatory diseases, ID is mainly the consequence of a disturbed iron homeostasis due to the release of inflammatory cytokines [4]. In this case, ID is called functional ID as it results from the sequestration of iron in otherwise quantitatively normal or abundant stores. ID can be also due to an insufficient iron intake or absorption or chronic blood loss, leading to a decrease in the total iron supply in the body. This quantitative decrease in iron stores defines absolute ID. Of note, in chronic inflammatory diseases, these two types of ID are not mutually exclusive and are often associated [5].

The clinical consequences of disturbed iron metabolism in patients with chronic inflammatory diseases are established, although they are frequently underrecognized. The symptoms and clinical signs due to ID are usually of moderate intensity and often neglected, especially since they are not very specific and can be confused with those of chronic underlying disease [6]. This is particularly true of the main symptom, fatigue [7]. However, ID has been also associated with more severe consequences, such as impaired physical function [4,8] and reduced quality of life [9,10]. ID can have a particularly severe impact in patients with chronic inflammatory conditions by aggravating the underlying medical condition and causing more rapid clinical deterioration. This has been evidenced in patients with chronic HF [11], non-dialysis CKD (ND-CKD) [12,13], IBD [14,15], and cancer [3]. Even if anemia is often the final consequence of ID [5], several studies showed that independently of the presence of anemia, ID appeared as a risk factor of mortality in patients with chronic HF (CHF) [11,16] and CKD [12,13]. Recently, we showed that the prevalence of ID is high in all patients with chronic inflammatory disease, independently of the presence of factors known to influence the occurrence of ID [17,18,19,20]. Therefore, ID should be considered an independent therapeutic objective in patients with chronic inflammatory conditions, that is, a comorbidity that should be systematically detected and treated if identified [4].

Under-diagnosis is frequent despite the high prevalence and potential deleterious consequences of ID in patients with a chronic medical condition. Under-diagnosis of ID can be related to a low recognition by the practitioners of ID as a potentially severe condition in patients with inflammatory diseases, but mainly to the absence of harmonized guidelines for ID diagnosis [21]. Different recommendations for monitoring iron status and diagnosis of ID exist according to the various professional associations worldwide [2,22].

Although the investigation of bone marrow iron stores is considered the gold standard, it cannot be used routinely. Of all the available biomarkers, circulating ferritin concentration and transferrin saturation (TSAT) index are commonly used in clinical practice and recommended for diagnosing ID in most international guidelines [2,5,6,23]. Ferritin reflects stored iron, and TSAT is indicative of transported iron. Always present, the decrease of TSAT is thus an essential diagnostic criterion of absolute or functional ID [24,25]. Since the reliability of serum ferritin measurement may be compromised in patients with chronic diseases or any other condition due to an inflammatory state, there is a large consensus in guidelines to determine serum ferritin and TSAT together in the first line. The heterogeneity between the different guidelines is mainly due to the recommended diagnostic thresholds and decision algorithms, which vary according to the disease under consideration. A consensus definition of ID, whatever the chronic condition, which is clinically relevant and easy to implement, could improve the detection of iron-deficient patients in daily clinical practice [2,23,26].

In this article, we assessed the prevalence of ID in patients with chronic inflammatory diseases using a common definition for the diagnosis of ID, whatever the underlying chronic disease. For that purpose, we used data collected during the CARENFER studies, which were nationwide cross-sectional surveys carried out in France in patients with cancer [17], HF [18], CKD [19], and IBD [20].

## 2. Materials and Methods

### 2.1. The CARENFER Studies

The CARENFER studies were conducted in France between May 2019 and June 2021. The four of them consisted of a cross-sectional, multicenter, and nationwide study carried out to determine the prevalence of ID in patients with cancer [17], HF [18], CKD [19], and IBD [20].

Briefly, for each study, the participating hospitals were selected based on a voluntary basis. All patients present in the disease’s specific wards during the study period, whether in-patient or out-patient, with the diagnosis of interest (solid tumor, chronic or decompensated HF, IBD according to international criteria, and non-dialysis CKD (stages 1 to 4)) were eligible. Few inclusion criteria were considered to limit selection bias: 18 years old or older, registration with a social security system, and written informed consent. Patients under guardianship or curatorship, as well as pregnant or breastfeeding women, were omitted.

For all included patients, a standardized questionnaire was conducted. The following information was retrieved from the patient’s medical record: patient’s demographic and clinical data; date, type, and reason of hospitalization; disease-specific characteristics; and ongoing treatment for ID.

For patients who had a recent (i.e., within 7 days before their inclusion in the present study) biochemical assessment, including Hb concentration, serum ferritin concentration, and TSAT index, no additional biochemical assessment was performed at inclusion.

The protocols complied with the Declaration of Helsinki recommendations, the International Conference on Harmonization (ICH) guidelines for good clinical practice (GCP), and all applicable laws, rules, and regulations. They also complied with the French laws and regulations. Ethical approvals were granted by an Ethical Committee (Comité de Protection des Personnes) designated by the French Ministry of Health. All subjects provided written informed consent. The ClinicalTrials.gov identifiers are NCT03924258, NCT03924271, NCT0412399, and NCT04123145.

### 2.2. Definitions and Statistics

First, we used official definitions of ID as established by scientific organizations related to each specific disease, that is, the European Society for Medical Oncology (ESMO) in 2018 [27], the European Society of Cardiology (ESC) in 2016 [28], the European Crohn’s and Colitis Organisation (ECCO) in 2015 [29], and the European Renal Best Practice (ERBP) in 2013 [30] and the Société Francophone de Néphrologie Dialyse et Transplantation (SFNDT) in 2020 [31] (Table 1). Then, we used a common definition of ID differing partly from the officially accepted definitions, where ID was defined as ferritin concentration <100 μg/L and/or a TSAT < 20%. This latter definition was based on the recommended diagnostic thresholds in the four considered pathologies, with available data showing that most patients with bone marrow ID have TSAT <20% and ferritin <200 μg/L [32], and clinical practice. It was discussed and agreed upon by the authors during a series of working group meetings in 2019.

Descriptive statistics included patients’ demographic and main clinical disease characteristics, as well as ongoing disease-specific treatment and completed iron therapy. Continuous variables with a Gaussian distribution are presented as mean ± standard deviation (SD). For variables with non-Gaussian distribution, the data are shown as medians with interquartile ranges (IQR). Normality was checked by the Shapiro–Wilk statistic. Categorical data were expressed as percentages.

In each study population (i.e., cancer, HF, IBD, and CKD patients), the prevalence of ID was estimated using both the officially accepted and the common definition of ID. In addition, the proportions of patients with a ferritin concentration <100 μg/L and a TSAT <20% were computed. All statistics were performed using SAS^®^ Version 9.4.

## 3. Results

### 3.1. Study Populations

The general and clinical characteristics of the patients included in the four CARENFER studies are presented in Table 2. A total of 1232, 1733, 1090, and 1245 patients with cancer, HF, IBD, and CKD were included between 9 May 2019 and 29 June 2021, respectively. Males were overrepresented in HF and CKD patients. In terms of severity, more than half of cancer patients (52.2%) received metastatic treatment, half of HF patients were NYHA III or IV, less than 10% of patients had a severe highly active IBD, and more than half of patients (53.0%) had a CKD at stages 3B or 4. Whatever the disease, a low proportion of patients were currently treated with—or had recently completed—an oral or intravenous iron therapy, except patients with IBD, who received intravenous therapy in more than 20% of cases.

### 3.2. Prevalence of ID According to Officially Accepted vs. Common Definition of ID

When using the definitions of the different medical societies (Table 1), the prevalence of ID in CARENFER studies was 57.9% in cancer patients, 49.6% in HF patients, 23.7% in IBD patients, and 47.1% in CKD patients (Figure 1).

A TSAT < 20% was 2 to 3 times more common than a serum ferritin concentration <100 μg/L in cancer and HF patients, arguing for functional ID. Similar proportions of TSAT < 20% and serum ferritin concentration <100 μg/L were reported in CKD patients. In contrast, among IBD patients, a serum ferritin concentration <100 μg/L was one and a half times more common than a TSAT < 20% (Figure 1).

When defining ID as the combination of a ferritin concentration <100 μg/L and/or a TSAT < 20%, ID prevalence estimates increased to 58.1% (vs. 57.9%), 62.8% (vs. 49.6%), and 61.2% (vs. 23.7%) in cancer, HF, and IBD patients, respectively. Since this common definition and the ERBP/SNFDT definitions are similar, the ID prevalence estimate was not modulated in CKD patients (47.1%).

The slight percentage difference in ID found in cancer patients when using the ESMO 2018 definition vs. the common definition (57.9% vs. 58.1%, respectively) was because patients with a TSAT < 20% but undetermined ferritin concentration could not be classified as iron-deficient according to ESMO guidelines.

The highest increases were observed in HF (+13.2% absolute difference) and IBD (+37.5% absolute difference) patients. In HF patients, the percentage difference in ID (49.6% vs. 62.8%) was mainly driven by patients with both TSAT < 20% and a ferritin concentration >300 μg/L; these patients represented more than 20% of the patients with a TSAT < 20% (Figure 2) and were considered as iron-deficient based on our common definition only. As depicted in Figure 2, in IBD patients, the high difference in ID prevalence when using the ECCO 2015 vs. the common definition (23.7% vs. 61.2%, respectively) was explained by more stringent thresholds for both TSAT and ferritin concentration with the ECCO definition.

## 4. Discussion

International recommendations for ID diagnosis, particularly in chronic inflammatory diseases, are very heterogeneous [2,23]. This is related to the fact that professional associations for each specialty have developed their own guidelines. To define ID, most guidelines recommend the measurement of both serum ferritin and TSAT, but with different diagnostic thresholds. ID is then defined as a standalone entity, independent of anemia, or defined in the case of concomitant anemia [27,33]. The high heterogeneity between guidelines contributes to their poor perception and adoption in clinical practice.

In this study, we used a common definition for the diagnosis of ID corresponding to ferritin concentration <100 μg/L and/or a TSAT < 20%. This definition was chosen on both scientific and operational grounds. It relates to the diagnosis of ID only. Although some laboratory markers of iron status might be available for ID diagnosis, it is well established that TSAT index is the most specific marker that correlates with the reduction of the total serum iron availability in chronic inflammatory conditions [5]. Ferritin is increased independently of iron status in acute and chronic inflammatory disorders, malignant disease, and liver disease. Since some patients may have CRP levels <5 mg/L, as observed for IBD patients, both ferritin and TSAT measurements were used in the common definition of ID. Cutoffs of 100 μg/L for ferritin and 20% for TSAT [2,5,23,34,35] result from studies using bone marrow as the gold standard [24] and clinical studies that correlate these biomarkers’ thresholds with patients’ symptoms [36,37] or improvement of these symptoms by an iron treatment [38,39].

By applying this common definition to an extensive panel of patients with cancer, HF, IBD, and CKD, a similar or higher prevalence of ID than with officially accepted definitions was found. In particular, it allowed the identification of 1.3 and 2.6 times more iron-deficient patients with HF and IBD, respectively. In addition, we confirmed that ferritin concentration alone was not sensitive enough, particularly in patients with cancer and HF, reaffirming the need to use both ferritin concentration and TSAT index in patients with chronic inflammatory diseases [23,32].

Several professional societies, including the Association Francophone pour les Soins Oncologiques de Support (AFSOS) [40], the ESMO [27], and the US National Comprehensive Cancer Network (NCCN) [41], have proposed criteria for the diagnosis of ID in cancer patients with anemia, differentiating between absolute and functional ID. In the absence of a consensus on cut-off values, these recommendations agree on one point, namely the importance of the joint determination of ferritin concentration and the TSAT index. Based on 2018 ESMO guidelines, ID is defined as either serum ferritin <100 μg/L (absolute ID) or the combination of serum ferritin ≥100 μg/L and TSAT < 20% (functional ID) [27]. By defining ID as ferritin <100 μg/L and/or TSAT < 20%, we were able to identify a slightly higher proportion of iron-deficient patients, including those with a single measurement of TSAT without ferritin concentration determination. Better detection of ID using this definition is warranted in all cancer patients, whether anemic or not, and regardless of the stage of the disease. We recently showed that the prevalence of ID in cancer patients receiving adjuvant or neo-adjuvant treatment was high and of the same magnitude as that reported in patients under metastatic treatment [17]. In addition, TSAT < 20% as the sole criterion for defining ID had a high sensitivity, justifying its use in combination with ferritin concentration to identify the actual population of iron-deficient patients.

The rationale for using an increased ferritin threshold (to 100 μg/L or even 300 μg/L) in HF is based on Nanas’ study [34], which showed that ferritin concentrations were normal. At the same time, ID was confirmed using a myelogram in most anemic patients with decompensated HF. This definition of ID in HF—ferritin concentration <100 g/L (absolute ID) or ferritin concentration 100–299 μg/L and TSAT < 20% (functional ID)—is often referred to as the “FAIR-HF criteria” because of the pivotal FAIR-HF trial [42]. It is now referred to as the “ESC 2016 criteria” [28,43]. Based on the results of epidemiological studies and clinical trials, the ESC issued a 2016 recommendation (grade I, level of evidence C) to look for ID—using both ferritin concentration and TSAT index—in all patients newly diagnosed with HF [28]. More recently, in 2021, ESC released new recommendations that “all patients with HF should be periodically screened for anemia and ID with a full blood count, serum ferritin concentration, and TSAT” [44]. By defining ID as ferritin <100 μg/L and/or TSAT < 20%, we were able to identify 26% more “iron-deficient” patients, who were mainly those with both TSAT < 20% and a ferritin concentration >300 μg/L. Among those patients, some of them might not require iron therapy but only close monitoring. Indeed, our definition applies to ID diagnosis only. Notably, ESC guidelines recommend to treat only patients with serum ferritin <300 μg/L and TSAT <20%, which prevents overtreatment [28]. Other thresholds or even biomarkers may be more relevant for identifying iron-deficient patients needing iron therapy. In HF patients with low left ventricular ejection fraction (LVEF), ID defined solely by TSAT < 20% vs. based on ESC 2016 criteria has been shown to be superior in the prediction of all-cause mortality [24,45]. Finally, our definition has the advantage of being applicable to all chronic inflammatory diseases, in particular CKD, which is frequently associated with HF [46].

Historically, professional societies and expert groups have issued inhomogeneous recommendations for ID diagnosis in CKD [2,4]. This is the case, for example, of the KDIGO (Kidney Disease: Improving Global Outcomes) [47] and ERBP [30] cooperative groups. However, the criteria are currently tending to be similar to those used in the field of HF. In particular, the expert consensus of the international IRON CORE Group [4], as well as the UK NICE group [48], propose the same definition as ours (i.e., ferritin concentration <100 μg/L or a TSAT < 20%), in both dialysis and non-dialysis patients. This definition has been endorsed by a group of experts from the SFNDT, which proposed an algorithm to help practitioners manage anemia and ID in patients with pre-end-stage kidney disease [31]. This group also recommends that iron assessment (based on both ferritin concentration and TSAT) should be part of the initial evaluation of all non-dialysis CKD patients and that ID (defined as ferritin concentration <100 μg/L or a TSAT < 20%) should be sought even in the absence of anemia, as ID is an independent risk factor of morbidity in this population [49].

In IBD, the criteria for ID diagnosis are controversial, with the choice of markers and thresholds depending on the concentration of inflammation associated with the IBD. Relatively old recommendations favor the interpretation of hematopoietic indices and ferritin concentrations [50]. The 2015 ECCO consensus proposes a definition that takes into account the presence or absence of associated inflammatory syndrome measured by CRP concentration: if CRP < 5 mg/L, absolute ID is defined as ferritin concentration <30 μg/L; if CRP ≥ 5 mg/L, functional ID is defined as ferritin concentration <100 μg/L [29]. In these guidelines, ID is likely underdiagnosed. Accordingly, an international expert consensus [4] has recently proposed an algorithm for managing IBD-associated ID defined as ferritin concentration <100 μg/L or TSAT < 20%, corresponding to the common definition we used in this study. Interestingly, when using this common definition for ID, more than half of patients (61.2%) with IBD presented with an ID, similarly to in the other studied inflammatory diseases. As depicted in Table 2, a higher proportion of IBD patients receiving a treatment for ID was reported (24.5% vs. 5.8% in cancer, 16.2% in HF and 11.9% in CKD patients). Most patients newly identified as iron-deficient using this common definition had an undetermined CRP concentration. Screening for ID based on ferritin concentration and TSAT, regardless of CRP concentration, would facilitate diagnosis in clinical practice. The existence of CRP positivity thresholds that vary according to whether or not ultra-sensitive CRP is measured also complicates the diagnosis of ID using CRP. Because inflammation can prevent intestinal absorption of iron from oral iron supplements, elevated CRP levels or “active disease status” are key in determining which treatment will be the most effective for these patients.

Using a universal and precise definition of ID in all chronic inflammatory diseases can improve screening for ID in clinical practice. While the French National Authority for Health (Haute Autorité de Santé, France) [51] recommends that diagnosis of ID in patients with inflammatory conditions must systematically be based on both serum ferritin concentration and TSAT, the precise definition and thresholds to be used are left to the discretion of the clinician. The vagueness of the definition certainly contributes to the low detection of ID in daily clinical practice. A large retrospective cohort study analyzed the use of ID markers prescription in all adult individuals who have had at least one iron treatment between 2006 and the end of 2015 using the French national medico-administrative database “Echantillon Généraliste des Bénéficiaires de l’Assurance Maladie (EGB)” [52]. Twelve percent of all individuals included in the EGB sample from the period 2006–2015 were prescribed iron replacement therapy. Approximately 25% of patients treated (24,000/97,000) were treated in the context of chronic inflammatory disease, including cancer and chronic HF. In those patients, a pre-treatment blood test was prescribed in only 34% and a post-treatment check-up in an average of 15% of cases. Similarly, a Veterans Affairs study found that more than one-third of patients with anemia and ulcerative colitis were not tested for iron-deficieny-associated anemia [53]. Moreover, among those who were diagnosed with iron-deficiency-associated anemia, 25% did not receive iron replacement therapy.

## 5. Conclusions

Adopting a common definition of ID for patients with chronic inflammatory conditions should contribute to better screening for ID, which is still insufficient, whatever the condition. This definition, combining ferritin <100 μg/L and/or TSAT < 20%, is consistent with the officially accepted definitions already recommended by most international professional associations. By applying this common definition, we aim to identify with a high degree of sensitivity, and in a more pragmatic way, patients who are iron deficient in order to propose to them an adapted follow-up, and if necessary, an iron therapy according to their clinical condition and associated comorbidities.

## Figures and Tables

**Figure 1 nutrients-14-01039-f001:**
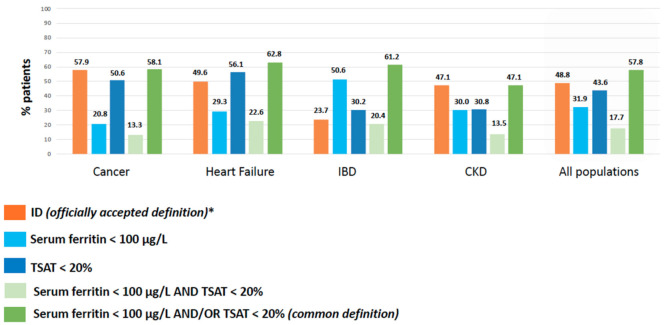
Prevalence of ID based on the officially accepted definition vs. common definition of ID in patients with cancer, heart failure (HF), inflammatory bowel disease (IBD), and non-dialysis chronic kidney disease (CKD). CARENFER studies [27,28,29,30,31]. ^∗^ As defined in CARENFER studies (i.e., based on ESMO 2018 [27], ESC 2016 [28], ECCO 2015 [29], and ERBP 2013 [30]/SFNDT 2020 [31] guidelines.

**Figure 2 nutrients-14-01039-f002:**
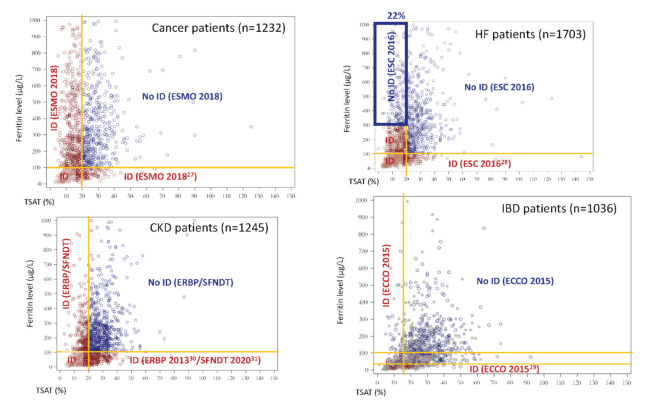
Distribution of ferritin concentration and TSAT index in the four types of CARENFER patients. Red dots represent iron-deficient patients based on ESMO 2018 guidelines (for cancer patients) [27], ESC 2016 guidelines (for HF patients) [28], ECCO 2015 guidelines (for IBD patients) [29], and ERBP 2013 [30] and SFNDT 2020 [31] recommendations (for non-dialysis CKD patients). Blue dots represent patients with no ID based on ESMO, ESC, ECCO, and ERBP/SFNDT definitions. Grey dots represent IBD patients with both undetermined CRP concentration and ID.

**Table 1 nutrients-14-01039-t001:** Definition of iron deficiency (ID) in CARENFER study patients with cancer, heart failure, chronic inflammatory bowel disease (IBD), or chronic kidney disease (CKD) according to both international or national guidelines and our common definition.

	International/National Guidelines		Common Definition
Cancer	ESMO 2018	Serum ferritin < 100 μg/L or Serum ferritin ≥ 100 μg/L and TSAT < 20%	Serum ferritin < 100 μg/L and/or TSAT < 20%
Heart Failure	ESC 2016	Serum ferritin < 100 μg/L or Serum ferritin (100 to 299) μg/L and TSAT < 20%	
IBD	ECCO 2015	Non-inflammatory context (CRP < 5 mg/L): Serum ferritin < 30 μg/L Inflammatory context (CRP ≥ 5 mg/L): Serum ferritin ≤ 100 μg/L	
CKD	ERBP 2013/SFNDT 2020	Serum ferritin < 100 μg/L and/or TSAT < 20%	
TSAT: iron-saturation of transferrin			
ESMO: European Society of Medical Oncology [27]			
ESC: European Society of Cardiology [28]			
ECCO: European Crohn’s and Colitis Organization [29]			
ERBP: European Renal Best Practice [30]			
SFNDT: Société Francophone de Néphrologie Dialyse et Transplantation [31]			

**Table 2 nutrients-14-01039-t002:** General and clinical characteristics of CARENFER study patients.

Characteristics	CARENFER Study Populations							
	Cancer		Heart Failure		IBD		CKD	
	N = 1221		N = 1661		N = 1036		N = 1211	
General characteristics								
Gender								
Male, n (%)	545	(44.6)	1023	(61.6)	496	(47.9)	737	(60.9)
Female, n (%)	676	(55.4)	638	(38.4)	540	(52.1)	474	(39.1)
Age (years), median (IQR)	64.0	(55.0; 71.0)	78.0	(76.0; 86.0)	39.0	(29.0; 53.0)	64.0	(51.0; 74.0)
BMI (kg/m^2^), median (SD)	24.4	(21.6; 27.8)	26.4	(23.0; 30.4)	23.9	(21.3; 27.4)	25.7	(22.7; 29.6)
Overweight/obesity, n (%) ^#^	548	(44.9)	972/1616	(60.1)	407/1021	(39.9)	597/1065	(56.0)
Disease characteristics								
Type								
Solid or hematological tumor ^&^	1221	(100.0)						
Acute decompensated HF *			887/1475	(60.1)				
Chronic HF			588/1475	(39.9)				
Crohn’s disease					685	(66.1)		
Ulcerative colitis					351	(33.9)		
Kidney transplantation							616/1211	(50.9)
Reason for admission								
Decompensation	NA		887/1475	(60.1)	44	(4.2)	NA	
Scheduled follow-up ^$^	NA		588/1475	(39.9)	992	(95.8)	NA	
Severity								
Cancer—Metastatic treatment	626/1199	(52.2)						
HF—NYHA III-IV			801/1601	(50.0)				
HF—LVEF < 40%			664/1502	(44.2)				
IBD—Disease remission					504/987	(51.1)		
IBD—Mild/Moderate activity					433/483	(89.6)		
CKD—Stage 3B and 4							640/1208	(53.0)
Time from disease diagnosis								
Median (IQR), in years	1.0	(0.0; 3.0)	NA		NA		5.9	(2.0; 14.7)
Ongoing treatment								
Disease-specific	1091	(89.4)						
Chemotherapy, n (%)	823/1091	(75.4)						
Proton pump inhibitors, n (%)			741/1660	(44.6)				
Aspirin, n (%)			566/1660	(34.1)				
Oral anticoagulant, n (%)			864/1660	(52.0)				
Anti-TNF					622/1035	(64.1)		
Immunosuppressive drug, n (%)					234/1035	(24.1)		
Current/completed treatment for ID								
Oral iron, n (%)	20/1215	(1.6)	81/1660	(4.9)	36	(3.5)	112	(9.2)
Intravenous iron, n (%)	49/1175	(4.2)	188/1660	(11.3)	218	(21.0)	33	(2.7)
Erythropoietin, n (%)	24/1213	(2.0)	NA		NA		241	(19.9)

HF: heart failure; IBD: inflammatory bowel disease; CKD: chronic kidney disease; NYHA: New York Heart Association; LVEF: left ventricular ejection fraction; BMI: body mass index; IQR: interquartile range; NA: Not applicable; ^#^ overweight and obesity were defined as a body mass index 25.0–30.0 kg/m^2^ and ≥30 kg/m^2^, respectively; ^&^ type of tumor not recorded; * patients with an acute decompensated HF corresponds to patients with acute or chronic HF with an unplanned hospitalization for decompensation; ^$^ scheduled visit or treatment initiation.

## Data Availability

The data that support the findings of this study are available on request from the corresponding author (P.C.). The data are not publicly available due to privacy or ethical restrictions.

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
