# Peer review of "Towards a Common Definition for the Diagnosis of Iron Deficiency in Chronic Inflammatory Diseases"

_nutrients, 2022, doi:10.3390/nu14051039_

Round 1

Reviewer 1 Report

This study aimed to investigate the prevalence of iron deficiency (ID) in patients with chronic inflammatory diseases using a common definition for the diagnosis of ID whatever the underlying chronic disease. Authors used data collected during the CARENFER studies, which were nationwide cross-sectional surveys carried out in France in patients with cancer, HF, CKD, and IBD. The need for some criteria to recognize ID regardless the underlying chronic disease is important in clinical practice in order to easily address the treatment. However, I believe that some points needed to be better explained/improved:

  • I believe that in IBD patients, it would be useful to consider whether the GI condition is in remission or active, measuring CRP levels or fecal calprotectin.
  • I believe that some other factors should be considered as i) women of reproductive age (for menstrual bleeding) ii) pregnant women.
  • It is not appropriate to cite unpublished paper (Cohen-Solal et al., submitted 2022, line 109; Peyrin-Biroulet et al., submitted 2021, line 110).
  • Line 108: It is not clear to me the design of the cited studies: they are cross-sectional or prospective studies?
  • Line 52: reference 1 is cited twice

Reviewer 2 Report

Cacoub et al. in their manuscript entitled "Towards a common definition for the diagnosis of iron deficiency in chronic inflammatory diseases" present a compelling study that suggests a 'common' definition of ID in chronic inflammatory disease. The topic is highly relevant at the clinic as ID in chronic inflammatory disease often remains underdiagnosed.    Major concern: 1. Figure 1. CKD graph: While the 'common definition' (dark green) shown in the graphs representing all the other diseases (Cancer, HF, IBD and All populations) is easy to interpret [the increase in ID prevalence is contributed by either serum ferritin (light blue) or TSAT (dark blue) percentages], it is not clear what is happening at the CKD graph: both ferritin and TSAT are having the same values, yet the % in the 'common definition' increased by 17%.    A detailed explanation would help.  Also, does it suggest that among the four conditions, CKD patients are likely to be benefited most from the 'common definition'?   2. It is appreciated that an inclusive and guideline would be useful for the screening purpose. The authors should also highlight it potential impact on overdiagnosis of ID and management strategies.    Minor concerns:  1. Abstract, Line 35: Please use the full term of TSAT.  2. Abstract Line 41: I think, the authors should rather use 'recommend' in place of 'confirmed' 3. Introduction, Line 72: Please include the abbreviation, and write as -  chronic HF (CHF) 4. Introduction, Line 94-95: Please include reference(s).       

Author Response

Please see teh attachment.

Round 2

Reviewer 1 Report

I believe that the current version of the manuscript is improved and it is suitable for publication